# PARP3 Affects Nucleosome Compaction Regulation

**DOI:** 10.3390/ijms24109042

**Published:** 2023-05-20

**Authors:** Alexander Ukraintsev, Mikhail Kutuzov, Ekaterina Belousova, Marie Joyeau, Victor Golyshev, Alexander Lomzov, Olga Lavrik

**Affiliations:** Institute of Chemical Biology and Fundamental Medicine, Siberian Branch of Russian Academy of Sciences, 630090 Novosibirsk, Russiakutuzov.mm@mail.ru (M.K.);

**Keywords:** nucleosome, atomic force microscopy, poly(ADP-ribose)polymerase, chromatin structure, DNA compaction

## Abstract

Genome compaction is one of the important subject areas for understanding the mechanisms regulating genes’ expression and DNA replication and repair. The basic unit of DNA compaction in the eukaryotic cell is the nucleosome. The main chromatin proteins responsible for DNA compaction have already been identified, but the regulation of chromatin architecture is still extensively studied. Several authors have shown an interaction of ARTD proteins with nucleosomes and proposed that there are changes in the nucleosomes’ structure as a result. In the ARTD family, only PARP1, PARP2, and PARP3 participate in the DNA damage response. Damaged DNA stimulates activation of these PARPs, which use NAD^+^ as a substrate. DNA repair and chromatin compaction need precise regulation with close coordination between them. In this work, we studied the interactions of these three PARPs with nucleosomes by atomic force microscopy, which is a powerful method allowing for direct measurements of geometric characteristics of single molecules. Using this method, we evaluated perturbations in the structure of single nucleosomes after the binding of a PARP. We demonstrated here that PARP3 significantly alters the geometry of nucleosomes, possibly indicating a new function of PARP3 in chromatin compaction regulation.

## 1. Introduction

DNA in eukaryotes is mostly packed into chromatin [1]. The compaction is implemented by chromatin proteins and is presumably regulated by modifications of nitrogenous bases in DNA or chromatin proteins. The compaction level can influence the expression of the affected genes via transcription regulation [2]. The basic unit of DNA compaction is the nucleosome. The main functions of nucleosomes are compaction and the protection of DNA and regulation of gene expression [3]. A nucleosome consists of 147 nt of DNA wrapped around a histone octamer consisting of two molecules of each of the following histones: H2A, H2B, H3, and H4. This nucleoprotein complex is also termed the nucleosome core particle (NCP). The structural and functional details are reviewed in Reference [3].

The higher compaction level is usually mediated by linker histone H1. This histone binds to an NCP in the entry–exit region, thus forming a chromatosome, and chromatosomes can then condense into fibers. This compaction of NCPs requires a certain density of DNA wrapping [4]. Parameters of the fiber may depend on the NCP compaction degree. For example, the replacement of the H3 histone with CenpA leads to less compacted NCPs [5,6]. This alteration probably results in an alternative type of NCP compaction in fibers [5]. Functions of different types of DNA compaction in chromatin are being debated. One of the known chromatin changes occurs in response to the binding of poly(ADP-ribose)polymerase 1 (PARP1) [7].

The diphtheria toxin-like ADP-ribosyltransferase (ARTD) family of proteins consists of 17 members. These proteins share the active site of the catalytic domain [8]. Three proteins from this family—PARP1, PARP2, and PARP3—are known to be DNA-damage-dependent. These PARPs activate in response to DNA damage. They catalyze the transfer of ADP-ribose from NAD^+^ to an acceptor molecule. Various proteins and DNAs can act as an acceptor for these PARPs [9,10,11]. PARP1 and PARP2 can synthesize long branched polymers of ADP-ribose (PAR), whereas PARP3 performs only mono(ADP-ribosyl)ation [12]. PARP1 and PARP2 are regulator proteins in base excision repair and double-strand break repair [13,14,15,16,17]. ADP-ribosylation can perform the function of an intracellular signal for the recruitment of DNA repair proteins. On the other hand, as a type of post-translational modification, ADP-ribosylation influences the properties of a target protein [18].

Several authors have revealed an interaction of the PARP1 protein with NCPs and have proposed that there is a change in the NCP structure as a result [19]. PARP1 can affect the chromatin structure via poly(ADP-ribosyl)ation (PARylation) [20]. Under PARylation conditions, PARP1 modifies histone H1, thus causing its dissociation. Recently, a protein involved in histone PARylation (HPF1) was discovered [21,22]. In the presence of this protein, PARP1 and PARP2 can modify (ADP-ribosyl)ate core histones in an NCP. These modifications lead to chromatin relaxation [23].

PARP1 can also directly affect NCPs. It has been shown that in the absence of linker regions, PARP1′s binding to an end of nucleosomal double-stranded DNA (dsDNA) causes a significant increase in the distance between adjacent gyres of the duplex, and this process is not accompanied by a loss of histones; moreover, it is reversible after PARylation [24]. Such major distortions of the NCP structure may be a consequence of the ability of PARP1 to strongly interact with DNA through the DNA-binding domain (DBD), which includes three Zn-finger domains, a WGR domain, and even a BRCT domain.

Although PARP2 is the closest homolog of PARP1, its DBD is considerably different. PARP2 does not contain any known DNA-binding motifs but comprises a structure similar to the SAP motif. It also has different DNA-binding properties: a lower affinity for free DNA and compacted DNA as compared to PARP1. It has been demonstrated that during the interaction with compacted DNA, PARP2 forms a bridge between two NCPs in double-strand breaks [25]. In contrast to PARP1, the interplay between PARP2 and an NCP has not been described in much detail.

In this regard, PARP3 is less characterized compared to PARP1 and PARP2, and the processes involving PARP3 are being researched at present. PARP3 interacts with PARP1 and several DNA damage repair proteins [26,27,28]. In the cell, PARP3 is reported to be associated with several polycomb group proteins [27]. The latter finding suggests that PARP3 participates in epigenetic regulation of transcription. Notably, this enzyme does not have a structurally separate DBD. The unstructured N-terminus is responsible for this function in PARP3. Nevertheless, PARP3 is strongly activated by DNA strand breaks in vitro and can facilitate non-homologous end joining [27,28,29].

More detailed information about aspects of structural reorganization during direct binding of a PARP protein to an NCP may be obtained by single-molecule methods such as atomic force microscopy (AFM). AFM is an approach used to directly measure the geometric characteristics of individual molecules placed on mica plaque surfaces. It is one of the most widely used nano-tools for studying protein DNA complexes including NCPs [30,31,32,33,34,35,36]. This method can be employed for estimating the NCP compaction degree by measurement of the angle between DNA arms near the entry–exit site of an NCP [31,37]. Using this approach, a stabilizing effect of histone H1 on an NCP has been demonstrated [32].

In our work, we studied the interactions of PARP1, PARP2, and PARP3 with an NCP reconstituted from native core histones and Widom’s clone 603 DNA extended by 79 and 120 bp DNA arms. In particular, we determined changes in the geometric parameters of NCPs during their binding to PARP1, PARP2, or PARP3.

## 2. Results and Discussion

### 2.1. The Localization of PARP Proteins in NCP–PARP Complexes

First, we determined the site of binding of each PARP protein to our model NCP. For this purpose, reconstituted NCPs were incubated with a PARP followed by immobilization on a mica surface and visualization by AFM scanning in air. Only images of complexes containing both PARP and NCP molecules were chosen for the analysis. According to the positioning of a PARP molecule, the captured images were sorted into two categories: (i) a PARP is located close to the NCP core; (ii) the PARP is located on the linker DNA region. Figure 1 shows typical images of NCPs in their complex with PARPs. While accumulating the data, we found that each of the three PARPs presumably binds near the NCP core: in 153 out of 200 complexes for PARP1, in 148 out of 200 complexes for PARP2, and 158 out of 200 complexes for PARP3.

Strong affinity (in the sub-nanomolar range) of PARP1 and PARP2 for DNA containing various structural elements has been demonstrated earlier [36,38]. In those experiments, naked DNA was used. Additionally, our previous data revealed that K_d_ values are almost identical when complexes of PARP1 with naked DNA and of PARP1 with an NCP are compared [39].

An earlier study uncovered a specific nature of PARP1′s binding to NCP and the ability of this protein to modulate chromatin structure through NAD^+^-dependent automodification without disassembly of the NCP core [40]. These authors also showed that PARP1 is associated with chromatin regions depleted of histone H1. PARP1 saturates chromatin in a molar ratio of 1:1 toward the NCP and competes with H1 for the binding to NCPs. A recent study shows the ability of PARP1 to bind DNA near the entry–exit site of an NCP through the BRCT domain of PARP1 in addition to Zn-finger domains [41]. Furthermore, a condensing effect of PARP1 binding on chromatin has been demonstrated [7]. These data are in agreement with our findings about PARP1 localization during its binding to the model NCP. Taken together, all these data may indicate a potential structural role of PARP1. The binding of PARP1 to an NCP instead of H1 in the absence of DNA damage may lead to a certain temporal pattern of chromatin alterations and to an alternative compaction degree.

Preferential binding of PARP2 to the NCP core was expected here because PARP2 possesses a significantly stronger affinity for NCP compared to naked DNA [39]. The mechanism underlying the interaction of PARP2 or PARP3 with the NCP is not clear, first of all, owing to dramatic differences in the structure of their DBDs from those of PARP1 and differences in subsequent various types of interaction with DNA [42,43]. Moreover, the interaction of PARP2 or PARP3 with the NCP in the absence of DNA damage may be mediated by core histones. In any case, the binding of PARP2 or PARP3 to an NCP may affect its geometry.

### 2.2. The Impact of PARP Binding on the NCP Compaction Degree

Here, we analyzed only the complexes where a PARP molecule is located close to the NCP core. We measured the angle between the linker DNAs of the NCP, i.e., the opening angle (as described in Materials and Methods), to evaluate the changes in the geometric parameters of the NCP. A similar approach was used previously [32,44].

We analyzed 200 complexes of the NCP under all conditions under study. As a reference sample in the experiment, we utilized an NCP without supplementation with any PARP. Graphical representation of the results is given in Figure 2b. The raw data are presented in Table A1, Table A2, Table A3 and Table A4. The average angle between DNA arms near the entry–exit region for NCPs in native states was estimated as 120° ± 5°. This result is consistent with the data obtained by Jan Lipfert’s group [31]. Those authors showed a dual-mode distribution in 2D density plots, which depicted a correlation between the length of unwrapped DNA and an opening-angle distribution. In contrast to their data, we did not observe such a clear-cut dual-mode distribution in our experiments (Figure A2). This discrepancy may be explained by a difference in the nucleotide sequences of the DNA used. In our work, we employed Widom’s clone 603 DNA (which is characterized by weaker affinity of binding to core histones) instead of clone 601 DNA as used in Reference [31]. The main difference between these two DNA sequences is the toughness of the NCP core: the NCP based on clone 601 DNA is tougher and therefore has less flexible DNA ends. It is probable that our model NCPs based on clone 603 DNA have insufficient differences in their opening-angle values to discriminate clearly between these two modes.

The binding of PARP1 to an NCP caused slight narrowing of the distribution of the opening arms’ angle without a significant effect on the compaction of the NCP (115° ± 4°). The difference in the measured values of the angle in the nucleosome in the presence and in the absence of PARP1 was not significant (even for a *p*-value of 0.9). As mentioned above, PARP1 can bind an NCP near the entry–exit site and interact with both DNA linkers. This interaction can influence the structural functioning of chromatin similarly to linker histone H1. It has been reported that the presence of H1 narrows the opening-angle distribution (meaning NCP stabilization) and does not change the compaction degree [32]. Furthermore, similarly to histone H1, PARP1 compacts chromatin, which is relaxed under PARylation conditions [7]. The authors of Reference [45] propose that the compaction is accomplished via the bringing of neighboring NCPs together by PARP1 molecules, analogously to the process observed in the polycomb group protein complex. This effect is probably due to loop formation caused by PARP1–PARP1 contacts [46]. It should be noted that the binding of PARP1 to DNA near the entry–exit site leads to the distancing of the two DNA gyres, thereby destabilizing the NCP core [24]. Thus, PARP1 loosens the NCP structure. Nonetheless, in that report, the authors demonstrated separation of fluorescent labels located on the DNA helices wrapping the histone core when PARP1 was bound. These data were obtained by the Forster resonance energy transfer technique, which does not discriminate between directions of the NCP deformation; the changes in NCP structure can occur in one of two directions: radial or axial. Because we did not detect significant changes in the compaction degree of the NCP in the presence of PARP1, the changes probably proceed in the axial direction. In this case, the influence cannot be determined by the method under study. We also cannot rule out that the previously described NCP structure distortions caused by PARP1 may affect only the DNA region that is in direct contact with the histone core. In this context, the geometry of the entry–exit site of DNA may be unaltered.

Even though PARP2 manifests significantly stronger affinity for NCPs than for naked DNA, PARP2 (just as PARP1) does not significantly affect the NCP compaction [39]. In the present work, neither the distribution of opening-angle values nor the compaction degree of the NCP was changed by the presence of PARP2 (121° ± 4°). Taking into account the standard deviation, the difference in the measured values of the angle in the nucleosome in the presence and in the absence of PARP2 was not significant (even for a *p*-value of 0.8). In the absence of blunt DNA ends, PARP2 probably binds to NCPs through histones. In this case, it is highly likely that PARP2 mostly binds outside the entry–exit site. Therefore, the impact on the compaction degree may be small.

Meanwhile, PARP3 exerted a distinctive effect on the compaction of the NCP core. We observed an increased compaction degree of the NCPs in the presence of PARP3 (104° ± 4°). Taking into account the standard deviation, the difference in the measured values of the angle in the nucleosome in the presence and in the absence of PARP3 was significant (a *p*-value of 0.001). Moreover, the presence of PARP3 induced the narrowing of the opening-angle distribution. It is worth mentioning that PARP3 is widespread in the nucleus as a part of polycomb group protein complexes. The molecular function of the polycomb group is important for homeotic gene regulation and is consequently suppressed during cell differentiation with the transition of genes into the heterochromatic state. Thus, the effect of PARP3 on NCP compaction may be required for the regulation of the access of other proteins to undamaged DNA via DNA compaction regulation.

In our work, we investigated changes in the NCP architecture during interactions with PARP1, PARP2, or PARP3 in the absence of (ADP-ribosyl)ation. The observed effects can be dramatically altered by the presence of lesions in DNA and NAD^+^. These alterations could also be important, especially because of the different abilities of PARP proteins to synthesize various PAR chains on an acceptor molecule, starting from the transfer of one ADP-ribose (as PARP3 does). What is more, the contribution of accompanying factors such as HPF1 could be substantial during the (ADP-ribosyl)ation and consequent NCP compaction reorganization.

Nevertheless, our study simulates the scenario where DNA is undamaged and the basic ADP-ribose transfer activity of PARPs is weak. To summarize, PARP3 is a new probable player in chromatin compaction regulation.

In conclusion, the clear difference between PARP1, PARP2, and PARP3 in their actions during this process may open up a new research field: the elucidation of PARP3′s function in chromatin compaction in the absence of DNA damage. The question is how to find the conditions (the biological process) where the observed effect is indispensable. The effect may be clarified when higher-order DNA compaction is studied in this context.

## 3. Materials and Methods

### 3.1. Reagents and Equipment

The following reagents and materials were used: 3.5 kDa cutoff dialysis membranes (Spectrum Laboratories Inc., Rancho Dominguez, CA, USA); bromophenol blue and xylene cyanol (Fluka, Buchs, Switzerland). Most of the reagents used in the study were purchased from Sigma (St. Louis, MO, USA). Recombinant Taq DNA polymerase was kindly provided by Prof. Svetlana Khodyreva (Institute of Chemical Biology and Fundamental Medicine, Siberian Branch of Russian Academy of Sciences (ICBFM SB RAS)). Recombinant proteins—human PARP1, murine PARP2, human PARP3, and histone octamers H2A, H2B, H3, and H4 from *G. gallus*—were prepared and isolated as described in References [11,47,48]. AFM imaging was performed on Multimode 8 (Bruker, Billerica, MA, USA) with the help of NSG30_SS probes (TipsNano, Tallinn, Estonia). The synthesis of 1-(3-aminopropyl)-silatrane (APS) was performed as described elsewhere [49]. NCP assembly products were visualized after separation in a polyacrylamide gel by means of a Typhoon FLA 9500 system (GE Healthcare Life Science, Barrington, IL, USA) and Amersham Imager 680 (GE Healthcare Life Science, Barrington, IL, USA).

### 3.2. Preparation of DNA Substrates

The DNA-603-containing substrate used in the experiments was generated by PCR from a pGEM-3z/603 plasmid vector (AddGene, Watertown, MA, USA) with unique primers. The DNA construct contains 147 bp of strong positioning of Widom’s clone 603 DNA sequence surrounded by plasmid DNA sequences of 120 and 79 bp:

5′-GGGCGAATTCGAGCTCGGTACCCGGGGATCCTCTAGAGTCGGGAGCTCGGAACACTATCCGACTGGCACCGAAACGGGTACCCCAGGGACTTGAAGTAATAAGGACGGAGGGCCTCTTTCAACATCGATGCACGGTGGTTAGCCTTGGATTGCGCTCTACCGTGCGCTAAGCGTACTTAGAAGCCCGAGTGACGACTTCACACGGTAGGTGGGCGCGCGAACTGGGCACCCGAGAGTGTCGATTATTTTACGGCTCACGCTGGGGTGATTTGTACTAGGAAAACGCCTATTCGTGTATTCCGCCTTGGTCATTAGGATCCCGGACCTGCAGGCATGCAAGCTTGAG-3′.

Primer oligonucleotides 5′-GGGCGAATTCNAGCTCGGTAC-3′ and 5′-CTCAAGCTTGCATGCCTGCAG-3′ were synthesized in the Laboratory of Biomedical Chemistry at the ICBFM SB RAS (Russia). The following program in PCR was used: 3 min at 94 °C; 30 cycles of 30 s at 94 °C, 20 s at 65 °C, and 1 min at 72 °C; with final extension for 3 min at 72 °C.

After the PCR-based synthesis, the DNA substrate was purified by gel electrophoresis and isolated from the gel by the protocol from reference [50].

### 3.3. NCP Assembly

The NCP assembly was carried out in accordance with our previously described protocol [51]. Briefly, by quick reconstitution of NCPs in analytical amounts, the correct ratio of DNA–histones’ species was determined. Then, preparative reconstitution was performed by gradient dialysis according to the determined ratio.

### 3.4. Preparation of NCP Samples Containing a PARP

Sample preparation for AFM imaging was performed as described before [52]. Freshly cleaved mica was functionalized with a solution of APS for sample deposition.

The reaction mixture was composed of 10 nM NCP, NCP buffer (20 mM NaCl, 0.2 mM EDTA, 1.6 mM CHAPS, 10 mM Tris-HCl pH 7.5, and 5 mM β-mercaptoethanol) and one of PARPs at a concentration of 10 nM (PARP1), 35 nM (PARP2), or 66 nM (PARP3). The reaction mixture was incubated for 15 min at 37 °C. Then, samples were diluted tenfold with Milli-Q water and immediately deposited on the mica surface. After 120 s of deposition, the mica surface was rinsed three times with 1 mL of Milli-Q water and dried in a gentle stream of argon. The samples were stored in a desiccator before the imaging.

### 3.5. AFM Imaging

The visualization was performed in tapping mode in air at a tip resonance frequency of 240–440 kHz. A typical resulting image had a size of 2 µm × 2 µm at 1024 pixels/row or 4 µm × 4 µm at 2048 pixels/row. The scanning rate was either 1.0 or 0.5 Hz, respectively.

### 3.6. Data Analysis

All images were first processed in the Gwyddion software (http://gwyddion.net/, accessed on 1 March 2023). The ImageJ software (https://imagej.nih.gov/ij/, accessed on 1 March 2023) was employed to measure parameters of the NCP core disk and of the NCP core in complex with PARPs, the length of the NCP DNA arm, and the angle between DNA arms. The arm length was estimated by measuring the DNA from the end point to the point of “entry” into the NCP disk. Diameters of the core and core PARP were estimated as the maximal distance between two parallel tangents to the disk. The angle between NCP DNA arms was defined as an angle formed by two beams from the center of the core disc to the “entry” points of DNA arms. On the basis of the obtained data, histograms and graphs were constructed using the SigmaPlot software v.11.0 (Systat Software Inc., Chicago, IL, USA). Variances in measured values were calculated by means of Student’s *t* distribution with 95% confidence intervals. Measured values are shown diagrammatically in Figure 2a. When sorting PARP–NCP complexes, we chose a distance of 3 nm between the NCP core and a PARP as the border point. The resolution of the cantilever used in this work allowed us to uniquely identify a PARP separated from the NCP core when the distance was more than 3 nm. Therefore, when the proteins were located closer to the NCP core, we assumed that they were directly interacting. The workflow is illustrated in Figure A1.

## Figures and Tables

**Figure 1 ijms-24-09042-f001:**
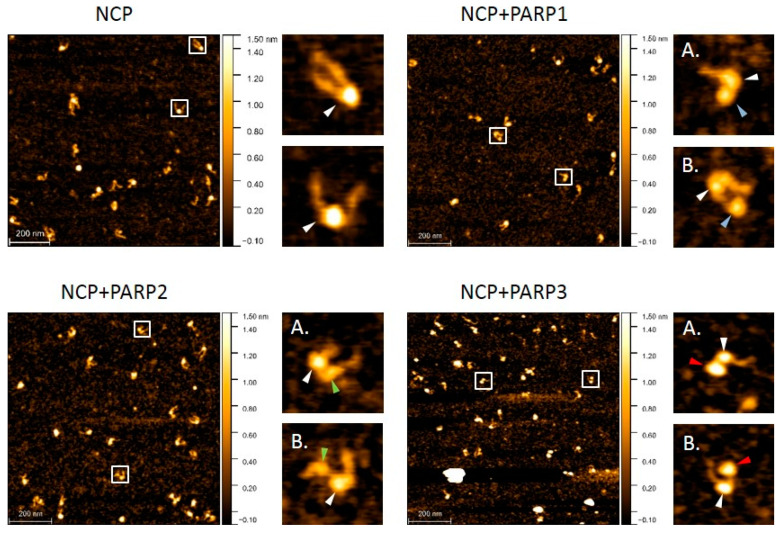
Representative AFM scans of an NCP. Cores of the NCP are indicated by white arrows. PARP1, PARP2, and PARP3 molecules are pointed out by blue, green, and red arrows, respectively. (**A**) A PARP located close to the NCP core. (**B**) A PARP located on the linker DNA region.

**Figure 2 ijms-24-09042-f002:**
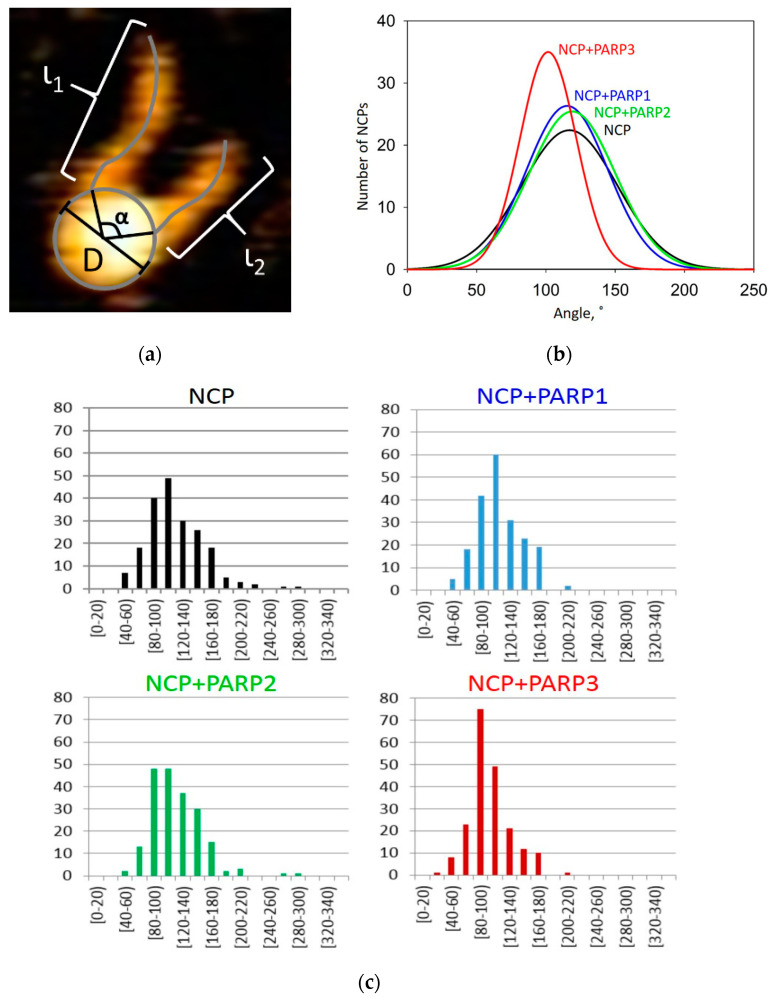
The compaction of NCPs depending on the presence of a PARP protein. (**a**) Schematic representation of determined parameters of an NCP. In the image, “l1” and “l2” are DNA arm lengths, “α” is the angle between DNA arms, and “D” is the diameter of the NCP core. (**b**) The Gauss interpolation of the distribution of “α” angle values. The black curve: NCP samples, the blue curve: samples of NCPs supplemented with PARP1, the green curve: samples of NCPs supplemented with PARP2, and the red curve: samples of NCPs supplemented with PARP3. (**c**) Representation of the distribution of “α” angle values. Black bars: NCP, blue bars: NCP supplemented with PARP1, green bars: NCP supplemented with PARP2, and red bars: NCP supplemented with PARP3.

## Data Availability

The data needed to reproduce our results are contained within the article. Raw data are available upon request.

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
