# Peer review of "PARP3 Affects Nucleosome Compaction Regulation"

_ijms, 2023, doi:10.3390/ijms24109042_

Round 1
Reviewer 1 Report
Authors have presented a significant research deciphering the significance of genome compaction which would definitely help our future researchers in developing potent therapeutic approaches. Abstract is well defined. Introduction is clear and well elaborated with sufficient citations. All sections are highly correlated with each other. Discussion is highly significant. However, i would suggest the potential author to reframe the title and give a strong futuristic conclusion. Altogether i would recommend the acceptance of this manuscript in its current form with minor suggestions mentioned above regarding title and conclusion.
Author Response
"Authors have presented a significant research deciphering the significance of genome compaction which would definitely help our future researchers in developing potent therapeutic approaches. Abstract is well defined. Introduction is clear and well elaborated with sufficient citations. All sections are highly correlated with each other. Discussion is highly significant. However, i would suggest the potential author to reframe the title and give a strong futuristic conclusion. Altogether i would recommend the acceptance of this manuscript in its current form with minor suggestions mentioned above regarding title and conclusion."
We thank the Reviewer for the careful reading of the manuscript and support of our work. We have changed the title of the manuscript and added some clarifications throughout the text that seemed to improve reading comprehension. We have also highlighted future directions of this research in a conclusion at the end of the Discussion section.
Reviewer 2 Report
In this manuscript, Ukraintsev et.al. utilized atomic force microscopy to study the possible interactions of PARP proteins with nucleosomes, and they concluded that PARP3 significantly alters the geometry of nucleosomes, but not the PARP1 and PARP2 without NAD+. Although interesting data have been shown, the main question of this study is lack of novelty. PARPs are famous DNA binding proteins and it is not a surprise that PARP3 alters the geometry nucleosomes. In addition, this manuscript was not well written and should be improved, especially for the abstract.
1. PARPs are highly associated with cellular apoptosis, any evidence to link the nucleosome binding features with their function?
2. The DNA sequence used in this study was from a plasmid vector, any sequence bias that may cause to PARPs binding? Any binding motifs for PARPs?
3. Statistical analysis should be added to figure.2
Author Response
"In this manuscript, Ukraintsev et.al. utilized atomic force microscopy to study the possible interactions of PARP proteins with nucleosomes, and they concluded that PARP3 significantly alters the geometry of nucleosomes, but not the PARP1 and PARP2 without NAD+. Although interesting data have been shown, the main question of this study is lack of novelty. PARPs are famous DNA binding proteins and it is not a surprise that PARP3 alters the geometry nucleosomes. In addition, this manuscript was not well written and should be improved, especially for the abstract."
Thanks to the Reviewer for the perusal of the manuscript and constructive comments.
Definitely, a sufficient number of studies have been devoted to the interaction of DNA with nuclear DNA-dependent proteins of the ARTD family: PARP1, PARP2, and PARP3. However, it should be noted that these proteins have considerable differences at the structural level from each other that require careful investigation of nucleic acid–protein interactions separately for each member of the family. First of all, compared to PARP2 and PARP3, the N-terminal part of PARP1 has several additional domains—specifically zinc-fingers and a BRCT domain—responsible for interaction with DNA [Karlberg et al., Mol. Aspects Med., 34 (2013), pp. 1088; Rudolph et al., Mol Cell. 81 (2021), 4994-5006.e5]. The binding of PARP1, PARP2, and PARP3 to a DNA duplex in the presence of NAD+ leads to the implementation of the catalytic activity of the proteins, the consequent synthesis of poly(ADP-ribose) by proteins PARP1 and PARP2, and the transfer of a mono(ADP-ribose) residue to the acceptor molecule by the PARP3 protein. It should be noted that upon DNA binding, all three PARPs don’t display any sequence specificity; however, the level of protein activation depends on the type of DNA structure. For instance, the maximal activation of PARP1 is observed during binding to a single-strand break, and for PARP2, during binding to a single-strand break containing a 5'-phosphate group [Ali et al., Nat Struct Mol Biol. 19 (2012), 685; Langelier et al., Nucleic Acids Res. 42 (2014), 7762]. DNA compaction affects the process of nucleic-acid binding by PARP proteins in the first instance due to the formation of a nucleosome core particle (NCP), which is a protein–nucleic acid complex with sufficient rigidity, limited flexibility, and restricted access to the nucleic acid. The interaction of the PARP proteins with the NCP should lead to a loss of contacts/partial reorganization of the protein–nucleic acid complex. In this respect, PARP1 is the most studied of the three proteins; some details have been described regarding the changes in the structure of a blunt-ended NCP during its interaction with PARP1 [Sultanov et al., AIMS Genet. 4 (2017), 21]. Despite the sufficient interest, the exact changes in the geometry of NCPs containing linker regions upon binding to PARP1 are unknown. As for PARP2 and PARP3, there are no such data at all. Therefore, the current work is a first-time study demonstrating the positioning of a PARP on DNA relative to the core particle as well as changes in geometric parameters of a nucleosome particle containing linker regions upon the binding of PARP1, PARP2, or PARP3. In addition, the results of the current work indicate an alternative type of interaction of PARP3 with an NCP. Our results are undoubtedly interesting in terms of a general model of interaction of the nuclear DNA-dependent proteins PARP1, PARP2, and PARP3 with compacted nuclear DNA in the absence of damage.
We have inserted appropriate explanations into the text (lines 63-84) and had the manuscript edited by an editing company.
- "PARPs are highly associated with cellular apoptosis, any evidence to link the nucleosome binding features with their function?"
Poly(ADP-ribosyl)ation plays an important role in cell death including apoptosis, necrosis, and a specific PAR-associated cell death pathway: parthanatos. It should be pointed out that up to 95% of cellular PAR is synthesized by PARP1. In this way, the PARP1 activation by DNA lesions leads to the formation of a PAR net intended to attract DNA repair machinery to restore the genome. At another point, the PARP1 activity results in high NAD+ consumption and subsequent depletion of ATP pools thus leading to passive necrotic cell death. On the other hand, being a substrate of activated caspases 3 and 7 in caspase-dependent apoptosis, PARP1 loses its activity after being cleaved, thereby suppressing DNA repair, and the cell dies by apoptosis. Moreover, in the presence of an excess of DNA lesions, PAR is translocated to the cytoplasm and turns on caspase-independent programmed cell death termed parthanatos, which is distinct from necrosis and apoptosis.
All these scenarios are driven and made noticeable by DNA lesions and contribute to PAR formation, i.e., are characterized by NAD+ involvement. Moreover, all these processes are not dependent on the cell cycle stage and, apparently, chromatin compaction degree. Our work addresses the situation where a PARP binds to an undamaged form of DNA (compacted or not); therefore, it’s not directly linked to any programmed cell death.
We cannot rule out a potential role of PARP1, PARP2, and PARP3 in cell death regulation during nucleosome binding. This may be a fruitful subject for studies in the future but is outside the scope of the present work.
- "The DNA sequence used in this study was from a plasmid vector, any sequence bias that may cause to PARPs binding? Any binding motifs for PARPs?"
In our work, for the construction of model NCPs, we used Widom’s DNA sequence “clone 603” with two linker regions. This sequence was cloned as is, without any codon optimization, and was amplified from a plasmid with unique primers and purified by extraction of a specific band after separation of the PCR products in the polyacrylamide gel. The accuracy of amplification was ensured by the selection of primers that do not have alternative binding sites in the plasmid. We have inserted appropriate explanations into the text (lines 246-249 and 265–262).
The DNA sequence does not contain any special PARP-binding motifs because PARPs have not been shown to be sequence specific yet. There are several reports about the specificity of PARP1, PARP2, and PARP3 in terms of binding to different DNA structures [Potaman et al., JMB, 348 (2005), 609, Obaji et al., 46 (2018) 12154, Langelier et al., NAR, 42 (2014) 7762].
- "Statistical analysis should be added to figure.2"
We added the statistical analysis to the main text of the manuscript (lines 144, 157, 184 and 191).
Reviewer 3 Report
Comments:
In this paper, the authors used atomic force microscopy (AFM) to investigate the roles of PARP1, PARP2 and PARP3 in related protein DNA complexes, namely nucleosome core particles (NCPs). Based on the observation, the author stated that it seems like the PARP3 significantly alters the geometry of nucleosomes, whereas the effects of PARP1 and PARP2 without NAD+ are negligible.
As a brief report, the introduction did a decent job of background explanation, and the result can be observed on the plot as well. But some mistakes remain in the texture and the whole structure of paper can be more compact and understandable by adding a workflow (illustration) chart to make the main contents extracted and presented clearly.
Questions:
1. Since multiple proteins, validations were used in this manuscript and most of them are served for different investigated purposes, the manuscript would be easier to access by adding a chart/illustration of the whole workflow at the start. Stating the purpose of each step and the brief result based on a certain pattern. A clear workflow would allow the reader to quickly catch up with the whole procedure, and also to have a clearer picture of the main contents.
2. The introduction did a decent job to explain the background knowledge of the topic. Each paragraph does its job of explaining the terms and meaning. Although it’s already a clean introduction, page 2 67 line, between the explanation of PARP1 and PARP2, then PARP3, I feel like adding a paragraph to briefly introduce the differences of these three proteins from the structural, functional and topological angles is necessary. And it would make the introduction more comprehensive.
3. For Figure 1, please circle the zoomed-out area in the bigger pictures to specify which part the smaller pictures presented.
4. For Figure 2B and 2C, although there are legends in the textures part, it is no harm to have the corresponding legends right on the plots, and it would serve the reader a better understanding.
5. For Figure A1,using a rainbow-colored 2D density scatter plot will give the reader a clearer result of wrapped length versus the opening-angle distribution of NCPs.
6. For table A4 (or A2), I believe the title is wrong, maybe it should be PARP3? “NCP parameters in presence of PARP3”?
7. Some minor grammar mistakes needed to be checked.

Author Response
"In this paper, the authors used atomic force microscopy (AFM) to investigate the roles of PARP1, PARP2 and PARP3 in related protein DNA complexes, namely nucleosome core particles (NCPs). Based on the observation, the author stated that it seems like the PARP3 significantly alters the geometry of nucleosomes, whereas the effects of PARP1 and PARP2 without NAD+ are negligible.
As a brief report, the introduction did a decent job of background explanation, and the result can be observed on the plot as well. But some mistakes remain in the texture and the whole structure of paper can be more compact and understandable by adding a workflow (illustration) chart to make the main contents extracted and presented clearly."
We are grateful to the Reviewer for the useful comments.
We have implemented the suggestions and added a figure to the Supplementary section illustrating the experimental design.
- "Since multiple proteins, validations were used in this manuscript and most of them are served for different investigated purposes, the manuscript would be easier to access by adding a chart/illustration of the whole workflow at the start. Stating the purpose of each step and the brief result based on a certain pattern. A clear workflow would allow the reader to quickly catch up with the whole procedure, and also to have a clearer picture of the main contents."
Indeed, this work deals with the study of the interaction of three DNA-activated proteins belonging to the ARTD family—PARP1, PARP2, and PARP3—with a nucleosome core particle (NCP). The main task was to determine the changes in the NCP geometry upon binding to a particular protein. In the final analysis, we did not aim to directly compare the behavior of these three enzymes with respect to a specific cellular process. Combining them into one experimental set is primarily explained by the ability of all three nuclear DNA-activated PARPs to interact with compacted genomic DNA. This property allows to use one method to characterize their protein–nucleic acid complexes. However, each of the PARPs potentially has its own role in the physiology of the cell, and these proteins should probably be considered independent elements of DNA metabolism.
As for the general experimental logic, we have added an explanatory figure to the Supplementary section.
- "The introduction did a decent job to explain the background knowledge of the topic. Each paragraph does its job of explaining the terms and meaning. Although it’s already a clean introduction, page 2 67 line, between the explanation of PARP1 and PARP2, then PARP3, I feel like adding a paragraph to briefly introduce the differences of these three proteins from the structural, functional and topological angles is necessary. And it would make the introduction more comprehensive."
Indeed, the members of the ARTD (PARP) family have significantly different structures and are united by a common structural hallmark, which is catalytic motif H-Y-[EDQ] [Luscher at al., FEBS J. doi.org/10.1111/febs.16142]. However, only three nuclear PARPs have the ability to be activated in response to DNA damage by DNA binding. PARP1 contains a DNA-binding domain (DBD) consisting of three Zn-finger motifs and WGR and BRCT domains. PARP2 is the closest homolog of PARP1 and differs mainly by the DBD, which does not contain canonical DNA-binding motifs but has a region with structure similar to the SAP motif. PARP3 lacks a structurally separate DBD; its unstructured N-terminal region is responsible for DNA binding. Thus, most of structural differences among these enzymes are determined by their DNA-binding regions which may be important for the comparison of their interaction with NCP attempted in this study.
We have added these explanations into the Introduction (lines 67-73 and 81-84).
- "For Figure 1, please circle the zoomed-out area in the bigger pictures to specify which part the smaller pictures presented."
We have added the necessary information.
- "For Figure 2B and 2C, although there are legends in the textures part, it is no harm to have the corresponding legends right on the plots, and it would serve the reader a better understanding."
Sorry to disagree, this information seems to be presented better in the figure legend, however we added several labels right on the plots.
- "For Figure A1,using a rainbow-colored 2D density scatter plot will give the reader a clearer result of wrapped length versus the opening-angle distribution of NCPs."
We have updated the corresponding figure according to the Reviewer’s comment.
- "For table A4 (or A2), I believe the title is wrong, maybe it should be PARP3? “NCP parameters in presence of PARP3”?"
We thank the Reviewer for such a careful reading of the manuscript. We have corrected the error.
- "Some minor grammar mistakes needed to be checked."
As requested, we have had the manuscript corrected by an English-language-editing service.
Round 2
Reviewer 2 Report
The authors cited several reports and tried to demonstrate the novelty of their study. However, their explanation does not release my concern without preforming additional experiments or analyses.
Specifically, PARP1 loses its activity after being cleaved does not mean that binding to an undamaged DNA is not directly linked to apoptosis. This requires additional evidence to make the conclusion (comment #1). For the DNA binding specificity, randomized DNA should be used instead of a fixed sequence to demonstrate the unbiased binding of PARPs (comment #2).
Author Response
"The authors cited several reports and tried to demonstrate the novelty of their study. However, their explanation does not release my concern without preforming additional experiments or analyses.
Specifically, PARP1 loses its activity after being cleaved does not mean that binding to an undamaged DNA is not directly linked to apoptosis. This requires additional evidence to make the conclusion (comment #1). For the DNA binding specificity, randomized DNA should be used instead of a fixed sequence to demonstrate the unbiased binding of PARPs (comment #2)."
We thank the Reviewer for the careful reading of our manuscript and sincere desire to improve it. We will definitely consider the possibility of studying structural changes in nucleosomes under conditions mimicking apoptotic ones with a focus on the regulatory role of defined ARTD family members in future research.
Indeed, PARP1 takes part in the processes of cell death in general and in apoptosis in particular. Moreover, ARTD family members participate in a list of metabolic processes involving chromatin organization, regulation of various stages of transcription, cell cycle control, signal transduction, DNA damage response and repair processes, mitochondrial homeostasis and regulation of oxidative stress response, protein stability through communication with the ubiquitination system, cellular transport, the immune system and the inflammatory processes regulation including the response to the viral agents' penetration, cell differentiation, programmed cell death, processes associated with the sleeping regulation, aging, and obesity, as well as cancer biology.
Nucleosome is the basic level of DNA compaction and our main interest is related to the study of potential changes in the compaction degree at this level. Various processes of the life-sustaining activity of a cell are directly related to one or another address to its genome, which is in a compacted state. In this regard, abundant nuclear proteins that regulate quite different metabolic processes can significantly contribute. In our model system, the primary goal was to determine the changes in the geometry of the nucleosome core particle upon binding with selected proteins of the ARTD family without reference to a specific biological process.
Relatively to the experiments with randomized DNA, we would like to clarify some details about nucleosome reconstitution. In eukaryotic cells, DNA is compacted in the nucleosomes in various parts of the genome. To date, no specific natural sequence has been found that is responsible for its preferential recognition by core histones and nucleosome formation. Consequently, this property leads to the irreproducibility of histone core positioning on DNA under nucleosome reconstitution in vitro. In this regard, for studying protein-nucleic acid interactions involving nucleosome the model sequences are used that provide greater stability and defined histone core positioning under the nucleosome reconstitution.
In this work, the nucleosome that reconstituted on the Widom 603 clone sequence was chosen for the study. It was selected by Widom’s group using a SELEX approach from sample a population of 5 × 1012 different chemically synthetic random DNA molecules, each having 220 bp of random DNA; their sequence was determined, and their free energy was measured. Widom and co-authors have carried out a variety of computational analyses to identify non-random features in the selected sequences [https://doi.org/10.1006/jmbi.1997.1494]. Thus, the sequence of clone 603 itself does not contain specific natural DNA sequences with respect to any functional activity for cellular metabolism, i.e., does not contain any information about the genes' sequencing, any regulatory elements, etc. This DNA is generally accepted basic sequence for the reconstruction of model nucleosomes.
As for the sequence specificity of PARP1 interaction with DNA, we have to note that to date no specific sequences have been found that ensure the interaction of PARP1 with DNA [doi: 10.1021/bi00439a050; D'Amours et al., Biochem J., 1999; doi.org/10.3390/ijms22105112]. At the same time, it is well known that the affinity of PARP1 for DNA may differ depending on the secondary structure of the DNA duplex; nevertheless, all the dissociation constant values are very close to each other and lie in the nanomolar range [doi: 10.1007/978-3-030-41283-8_4]. Multiple data are available for details of PARP1 activation; in general, they indicate that the affinity of PAPR1 for DNA and its activation degree do not correlate with each other [https://doi.org/10.1074/jbc.M708558200; https://doi.org/10.1093/nar/gku474; DOI: 10.1126/science.aax6367; DOI: 10.1126/sciadv.abq0414]. PARP1 activation also takes part in cell death regulation. The generally accepted theory is the promotion of the processes during the interaction of PARP1 with damaged DNA and subsequent activation of the protein, which manifests in the immediate synthesis of poly(ADP-ribose) [DOI: 10.1074/ jbc.RA120.014479]. Notably, PARP1 does not display specificity to DNA sequence here, only to the damage occurring. In this work, we used undamaged DNA in order to simulate the conditions of regular DNA compaction without nucleosome disturbing factors and we conduct all the experiments in the absence of NAD+. Thus, our data are related to the potentially consider possible alternative pathways for the regulation of the life-sustaining activity of a cell in the absence of damage.
As for the novelty of data in our manuscript, we would like to accentuate that this work showed the positioning of PARP2 and PARP3 proteins during interaction with the core particle for the first time. It also has been shown for the first time that PARP3 interaction can affect nucleosome compaction. As for PARP1, the results presented in this work complement previously known data on compaction with new details about the anisotropic effect of the protein on the geometry of the nucleosome core particle, which can lead to various consequences for cell metabolism.
In conclusion, we would like to express once again our gratitude to the Reviewer and we hope that the above theses will help to clarify his mentioned points of the study that could remain unclear. We believe that adding appropriate additional discussions to the main text of the manuscript will be redundant and complicate the perception of the main idea.